# Upgrading the Baryonic Matter at the Nuclotron Experiment at NICA for Studies of Dense Nuclear Matter

Peter Senger [1,2,*], Dmitrii Dementev [3], Johann Heuser [1], Mikhail Kapishin [3], Evgeny Lavrik [4], Yuri Murin [3], Anna Maksymchuk [3], Hans Rudolf Schmidt [1,5], Christian Schmidt [1] 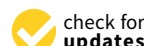, Anna Senger [1] and Alexander Zinchenko [3]

[1] GSI Helmholtzzentrum für Schwerionenforschung, 64291 Darmstadt, Germany; j.heuser@gsi.de (J.H.); h.r.schmidt@gsi.de (H.R.S.); C.J.Schmidt@gsi.de (C.S.); a.senger@gsi.de (A.S.)
[2] Institute of Nuclear Physics and Engineering (INPhE), National Research Nuclear University MEPhI, 115409 Moscow, Russia
[3] Joint Institute for Nuclear Research, 141980 Dubna, Russia; dementiev@jinr.ru (D.D.); kapishin@jinr.ru (M.K.); murin@jinr.ru (Y.M.); anna_maksymchuk@mail.ru (A.M.); Alexander.Zinchenko@jinr.ru (A.Z.)
[4] Facility for Antiproton and Ion Research, 64291 Darmstadt, Germany; e.lavrik@gsi.de
[5] Physics Institute, University of Tuebingen, 72076 Tuebingen, Germany
[*] Correspondence: p.senger@gsi.de

**Abstract:** The Nuclotron at the Joint Institute for Nuclear Research in Dubna can deliver gold beams with kinetic energies between 2 and 4.5 A GeV. In heavy-ion collisions at these energies, it is expected that the nuclear fireball will be compressed by up to approximately four times the saturation density. This offers the opportunity to study the high-density equation-of-state (EOS) of nuclear matter in the laboratory, which is needed for our understanding of the structure of neutron stars and the dynamics of neutron star mergers. The Baryonic Matter at the Nuclotron (BM@N) experiment will be upgraded to perform multi-differential measurements of hadrons including (multi-) strange hyperons, which are promising probes of the high-density EOS, and of new phases of quantum chromodynamic (QCD) matter. The layout of the upgraded BM@N experiment and the results of feasibility studies are presented.

**Keywords:** heavy-ion collisions; new experimental facilities; nuclear matter equation-of-state

## 1. Introduction

The experimental exploration of the high-density nuclear matter equation of state is an important goal for contemporary and future research, both in the laboratory and for astronomical observations. Heavy-ion beams provided by the future accelerator centers, the Facility for Antiproton and Ion Research (FAIR) in Darmstadt, Germany [1], and the Nuclotron-based Ion Collider Facility (NICA) at the Joint Institute for Nuclear Research (JINR) in Russia [2], offer unique possibilities to produce baryonic matter at high densities. The fixed-target Compressed Baryonic Matter (CBM) experiment at the FAIR [3] and the Multi-Purpose Detector (MPD) at the NICA collider [4] are designed to measure diagnostic probes, which will provide new information on the quantum chromodynamic (QCD) phase diagram at large baryon-chemical potentials, including the high-density equation-of-state (EOS). As an intermediate step, the Baryonic Matter at the Nuclotron (BM@N) fixed-target experiment has been installed at the Nuclotron at the JINR [5]. The current version of the BM@N experiment is well suited to perform experiments with beams of light ions. To measure Au + Au collisions at beam energies up to 4.5 A GeV and with reaction rates up to 20 kHz, the experiment will be upgraded with an evacuated

beam pipe, with a hybrid tracking system comprising of 4 stations of double-sided microstrip silicon sensors and 7 planes of two-coordinate gaseous electron multiplier (GEM) detectors (already partly existing), and with a zero degree calorimeter for event characterization. The silicon tracking detector system is presently under development in close cooperation with the silicon detector group of the CBM collaboration. This article will briefly outline the envisaged physics program of the upgraded BM@N experiment and the first results of the physics performance simulations.

The Nuclotron beam energies are well suited to produce dense nuclear matter in the collision of heavy nuclei. This is illustrated in Figure 1, which depicts the central density in the reaction volume of a central Au + Au collision at a beam energy of 5 A GeV as a function of time, as predicted by several transport models and a hydro-dynamical calculation [6]. According to these calculations, the nuclear fireball will be compressed to about 5 times the saturation density $\rho_o$ at a beam kinetic energy of 5 A GeV. Therefore, experiments using the Nuclotron beam allow to study the EOS at neutron star core densities. Depending on the EOS, the central densities of neutron stars with about two solar masses vary between 4 and 8 $\rho_o$ [7]. The central densities increase for neutron star models with smaller maximum mass, i.e., for softer equations of state.

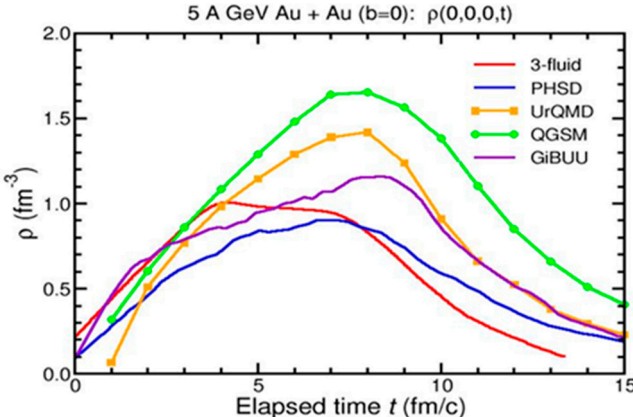

**Figure 1.** Net-baryon density versus time reached in a central Au + Au collision at 5 A GeV beam energy according to various transport codes and a hydro-dynamical calculation [6].

## 2. The High-Density Nuclear-Matter Equation-of-State

A fundamental ingredient for our understanding of nuclear matter is its equation-of-state (EOS), which determines the evolution of supernova explosions, the size and radius of neutron stars, and the dynamics of neutron star mergers. The EOS describes the relationship between density, pressure, volume, temperature, energy, and isospin asymmetry, and can be expressed as the energy per nucleon as a function of density:

$$E_A(\rho,\delta) = E_A(\rho,0) + E_{sym}(\rho)\cdot\delta^2 + O(\delta^4)$$

with the isospin asymmetry parameter $\delta = (\rho_n - \rho_p)/\rho$. Symmetric matter is stable around saturation density $\rho_o$ with a binding energy of $E/A(\rho_o) = -16$ MeV at saturation density $\rho_o$, the slope $\delta(E/A)(\rho_o)/\delta\rho = 0$, and the curvature $K_{nm} = 9\rho^2\delta^2(E/A)/\delta\rho^2$ with $K_{nm}$ the nuclear incompressibility. The EOS for symmetric nuclear matter is shown in Figure 2 for soft and hard Skyrme forces, and from microscopic ab-initio calculations [8].

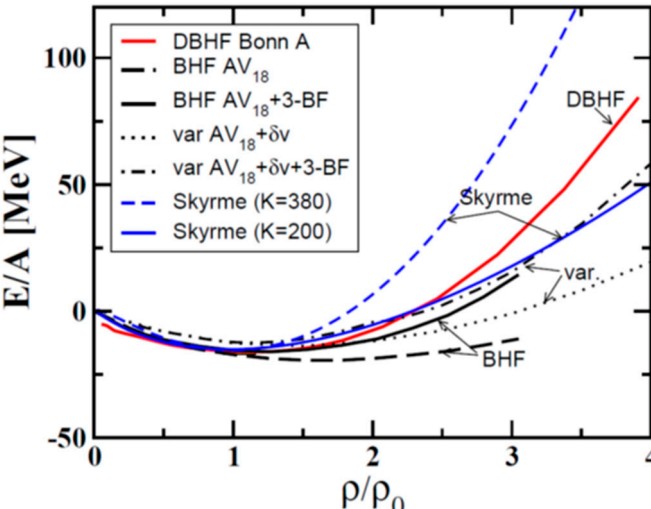

**Figure 2.** The equation of state for symmetric nuclear matter for soft and hard Skyrme forces and from microscopic ab-initio calculations [8].

Experiments at GSI have investigated the EOS of symmetric matter up to about 2 $\rho_o$ by measuring the excitation of $K^+$ meson production in Au + Au and C + C collisions at sub-threshold beam energies from 0.8 and 1.5 A GeV [9]. Sub-threshold kaon production proceeds via multiple collisions involving pions and Delta resonances. At high densities, which are reached if the EOS is soft, multiple collisions occur more frequently, and the kaon yield is enhanced. The measured data can be reproduced by transport model calculations when taking into account momentum-dependent interactions and a soft EOS with a nuclear incompressibility of about 200 MeV [8,10]. The data on the elliptic flow of protons, deuterons, tritons, and $^3$He in Au + Au collisions at beam kinetic energies between 0.4 and 1.5 A GeV have been explained by the isospin-dependent quantum molecular dynamics (IQMD) transport calculations when taking into account momentum-dependent interactions and assuming a soft EOS for symmetric matter, with nuclear incompressibility of $K_{nm}$ = 190 ± 30 MeV [11]. The EOS with hard Skyrme forces, i.e., with nuclear incompressibility of $K_{nm}$ = 380 MeV, as shown in Figure 2 is in contradiction with these results.

The BM@N experiment will explore the EOS of symmetric matter up to about 4 times the saturation density by measuring the elliptic flow of protons, which is driven by the pressure gradient in the fireball and, therefore, it is sensitive to the EOS. Pioneering experiments at the Alternating Gradient Synchrotron (AGS) in Brookhaven have already measured both the directed and the elliptic flow of protons in Au + Au collisions at beam kinetic energies between 2 and 11 A GeV [12]. However, the interpretation of these flow data using a transport model was not conclusive, in particular in the Nuclotron energy range. The data on the directed flow were compatible with a soft EOS, while elliptic flow data indicate a hard EOS [13]. Therefore, the aim of the BM@N experiment is to repeat these measurements with improved statistics and precision.

Another very promising observable that is sensitive to the high-density EOS of symmetric matter is the excitation function of multi-strange (anti-)hyperons. According to transport models, $\Xi$ and $\Omega$ hyperons are produced in sequential collisions involving kaons and lambdas and, therefore, are sensitive to the density of the fireball [14,15]. This sensitivity is expected to increase at beam energies close to or even below the production threshold. For example, the production of a Xi hyperon in a proton–proton collision pp $\rightarrow \Xi^-$ K$^+$K$^+$p requires a minimum proton energy of E$_{thr}$ = 3.7 GeV. Nevertheless, Xi hyperons can be produced in heavy-collisions via strangeness exchange reaction at energies above the lambda threshold of 1.6 GeV via lambda–lambda collisions $\Lambda^0\Lambda^0 \rightarrow \Xi^-$ p. Even Omega hyperons can be produced, when the Xi collides with another lambda $\Lambda^0 \Xi^- \rightarrow \Omega^-$ n or with a kaon K$^-\Xi^- \rightarrow \Omega^- \pi^-$. Therefore, the measurement of multi-strange hyperons will be a central goal of the BM@N research program at the Nuclotron.

## 3. Exploring Mixed Phases of QCD Matter

According to chiral mean field model calculations, a transition from nuclear to quark matter is expected during the evolution of a neutron star merger with a total mass of 2.8 solar masses, when a density between 3 and 4 $\rho_o$ and at a temperature of about 50 MeV is reached [16]. Such densities, although at higher temperatures, can be created in Au + Au collisions at Nuclotron energies. A promising observable, which is expected to be sensitive to the onset of such a phase transition, is the yield of multi-strange hyperons. In heavy-ion collisions at ultra-relativistic energies, the yield of (anti-) hadrons including multi-strange (anti-) hyperons and light nuclei can be perfectly described by the statistical hadronization model, which assumes chemical equilibrium [17]. In view of the fact, that the hyperon–nucleon scattering cross-section is small, this observation was interpreted as an indication that the system had undergone a transition from a partonic phase to the hadronic final state, with the equilibration of multi-strange hyperons being driven by multi-body collisions in the high-particle density regime near the phase boundary [18]. Agreement of the hyperon yield with thermal model calculations was also found at a beam kinetic energy of 40 A GeV in Pb + Pb collisions at the SPS [19]. The particle yields measured in Ar + KCl collisions at a beam kinetic energy of 1.76 A GeV can also be explained by the thermal model, except for $\Xi^-$ hyperons. The yield of $\Xi^-$ hyperons exceeds the thermal model prediction by about a factor 24 ± 9 [20], indicating that these particles are far off the chemical equilibrium. Therefore, the measurement of multi-strange hyperons in Au + Au collisions at different Nuclotron beam energies will explore the onset of equilibration of multi-strange hyperons at high net-baryon densities.

## 4. The BM@N Experiment

The layout of the proposed BM@N configuration is shown in Figure 3. The tracks of charged particles will be determined with two detector systems: 4 stations of double-sided microstrip silicon sensors downstream the target, and a set of 7 planes of two-coordinate GEM (gaseous electron multiplier) detectors mounted downstream of the silicon stations. Both the silicon tracking system (STS) and the GEM stations will be operated in the magnetic field (maximum value 1.2 T) of a large aperture dipole magnet with a gap of 1 m. The silicon detector stations are currently under development for the CBM experiment [21]. The tracking system behind the magnet actually consists of 2 large drift chambers (DCH), which will be replaced by cathode strip chambers for track measurement in Au + Au collisions. For particle identification, the time-of-flight will be measured with 3 walls of resistive plate chambers (mRPC) with strip readout. The T0 detector is positioned around the target and will provide the start signal for the time-of-flight (TOF) measurements together with the trigger signal for the data acquisition. The total energy of the projectile spectator fragments will be measured by a zero degree calorimeter (ZDC) to determine the centrality of the collision and the orientation of the reaction plane. The existing BM@N detector systems are described in [22].

### 4.1. Radiation Calculations

The current version of the BM@N experiment is used to measure reactions with low-intensity beams of light ions, which traverse the setup in the air without any beam pipe. For future measurements with a high-intensity gold beam, the setup has to be upgraded with an evacuated beam pipe. Such a beam pipe has been designed, consisting of carbon fiber tubes of 1 mm wall thickness with increasing diameter and several kinks to follow the deflection of the beam in the magnetic field. This design has been validated by FLUKA [23] calculations assuming a gold beam with a kinetic energy of 4.5 A GeV, a profile with a width of $\sigma$ = 3.5 mm, a divergence of 1 mrad, and an intensity of $2 \times 10^6$ ions/s, hitting a gold target with a thickness of 250 μm located at $Z = 0$. The resulting distribution of the gold ions projected in the *X–Z* plane is shown in Figure 4. The calculations demonstrate, that the curvature and the increasing diameter of the proposed beam pipe are well suited to guide the high-intensity beam and to minimize radiation outside the pipe. The FLUKA calculation takes into account the divergence

of the beam, the multiple scattering of the gold ions in the target, and the reactions of the beam in the material of the beam pipe. The particles produced in the target including the delta-electrons are not shown in Figure 4.

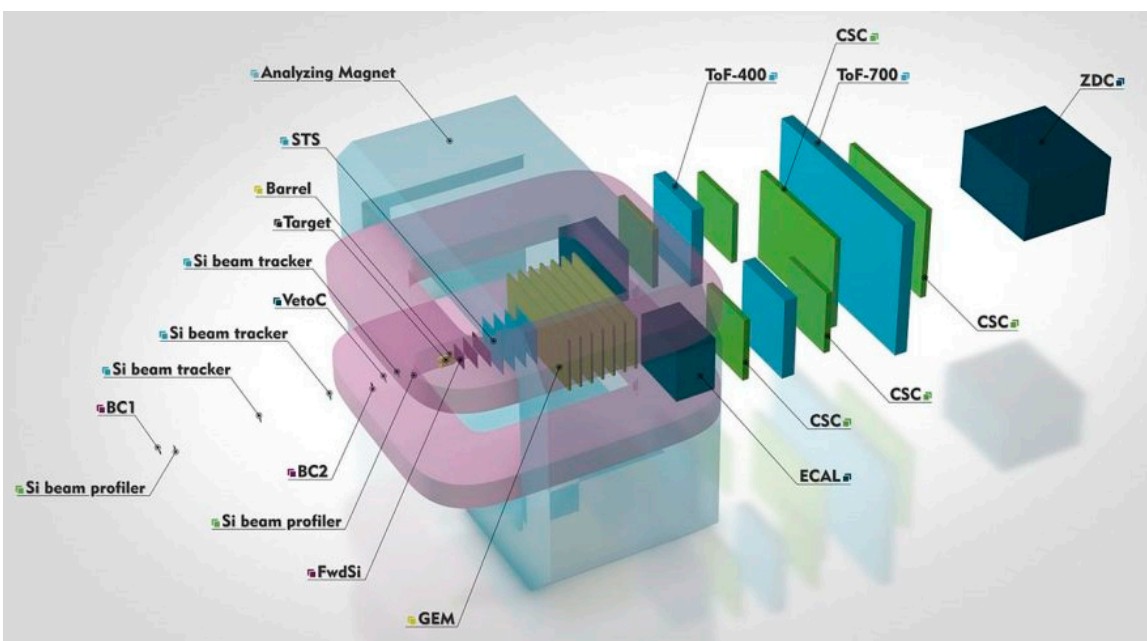

**Figure 3.** Sketch of the Baryonic Matter at the Nuclotron (BM@N) experiment with its various detector systems.

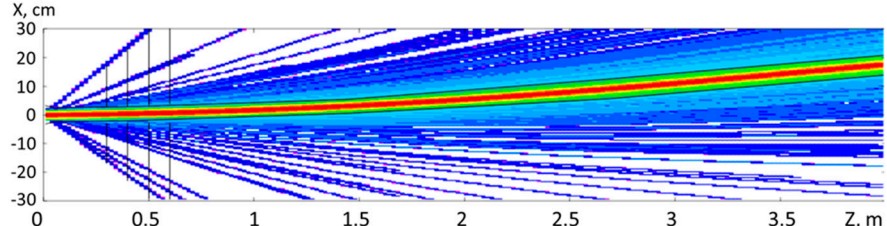

**Figure 4.** Projection of the beam particles in the *X–Z* plane illustrating the beam deflection as calculated with the FLUKA code for an Au beam with a kinetic energy of 4.5 A GeV, a profile with a width of σ = 3.5 mm, and a divergence of 1 mrad.

In the region of the silicon stations, the ionizing dose reaches values of about 10 Gy after 2 months of beam on target. Assuming a total running time of the BM@N experiment of 5 years and 4 months per year, a dose of about 100 Gy is reached, resulting in mild damage to the sensors in the inner part of the stations. The equivalent neutron fluence in the silicon stations reaches values below $10^{10}$ $n_{eq}/cm^2$ after 2 months, corresponding to a lifetime fluence of $10^{11}$ $n_{eq}/cm^2$, which is well within the radiation tolerance of the sensors.

In the region of the GEM stations, the ionizing dose reaches values of about 1 Gy after 2 months of beam on target, corresponding to a lifetime dose of 10 Gy. The equivalent neutron fluence in the GEM stations also reaches values below $10^{10}$ $n_{eq}/cm^2$ after 2 months, corresponding to a lifetime fluence of $10^{11}$ $n_{eq}/cm^2$. Both values can be tolerated by the GEM detectors.

*4.2. Physics Performance Simulations of the BM@N Hybrid Tracking System*

As a first step, the lambda hyperon reconstruction performance of the BM@N hybrid tracking system has been studied based on 1000 central Au + Au collisions with a beam kinetic energy of 4 A GeV produced with the DCM-QGSM event generator [24]. For track reconstruction, the latest version

of the CBM Cellular Automaton (CA) tracking code was used [25]. The BM@N hybrid tracking system comprises of 4 silicon stations and 6 GEM stations. The silicon stations are located at 30, 50, 70, and 90 cm downstream of the target, followed by GEM detectors at 120 cm up to 270 cm, with a gap of 30 cm between the stations.

The reconstruction efficiency for primary tracks as a function of momentum is presented in the left panel of Figure 5. The primary tracks, which create only hits in the 4 silicon stations, can be reconstructed with an efficiency of better than 90% above a momentum of about 0.6 GeV/c, whereas the track reconstruction efficiency for the hybrid system reaches a maximum of about 90% between 1 and 2.5 GeV/c, and then drops to about 80% at 6 GeV/c. The reason for the lower efficiency of the Silicon + GEM system is the low granularity of the GEMs, which leads to a large number of clone hits being misinterpreted as real hits. The momentum resolution for primary particles as a function of momentum is shown in the right panel of Figure 5 for the hybrid tracker. Only for momenta below $p = 0.5$ GeV/c, the momentum resolution is worse than $\Delta p/p = 0.006$.

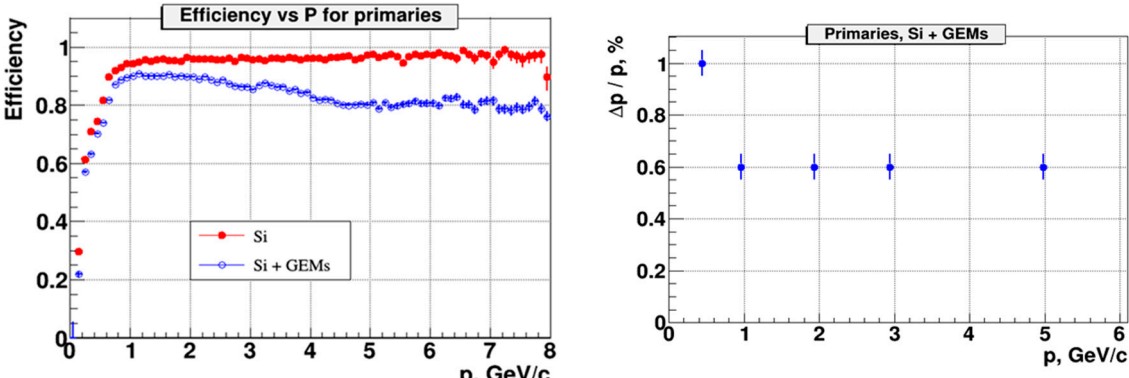

**Figure 5. Left panel**: Reconstruction efficiency as a function of momentum for primary tracks with a minimum of 4 hits in the Si stations only (**red histogram**), and in the Si + GEM stations (**blue histogram**). **Right panel**: Momentum resolution for primary tracks emitted in the central Au + Au collisions at a beam kinetic energy of 4 A GeV reconstructed in the BM@N hybrid tracking system as a function of momentum.

Lambda hyperons are reconstructed by the cellular automaton (CA) track finder by identifying the vertex of the lambda decay into two charged particles ($\Lambda \rightarrow p + \pi^-$), assuming that the decay products are either a pion or a proton, and vice versa. It is worthwhile to note, that no particle identification of the lambda decay products has been applied. The resulting proton–pion invariant mass spectrum is shown in Figure 6. The mass resolution of the lambdas is $\sigma = 1.1$ MeV/c$^2$, the signal-to-background ratio is S/B = 5.2.

The phase space distributions of lambdas and their decay products emitted in central Au + Au collisions at a beam kinetic energy of 4 A GeV as generated with the DCM-QGSM code are shown in the upper row of Figure 7 (lambdas left, protons center, pions right). The proton and pion identification has been performed based on Monte Carlo information. The lower row depicts the distribution of lambdas (left), decay protons (center), and decay pions (right) reconstructed by the hybrid tracking system. According to Figure 7, about 3800 lambdas are produced in 1000 central Au + Au collisions at 4 A GeV kinetic beam energy. The number of the decay products correspond to the branching ratio of the lambda decay $\Lambda \rightarrow p + \pi^-$, which is BR = 0.64. The lambda reconstruction efficiency is slightly above 10% for the hybrid detector system.

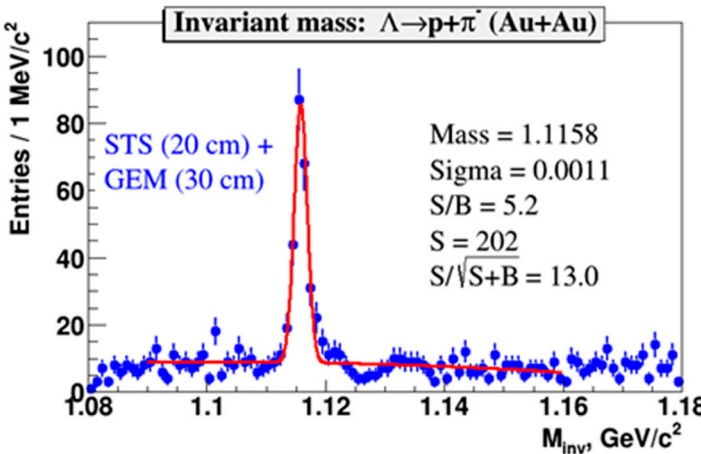

**Figure 6.** Proton–pion invariant mass spectra using the BM@N hybrid tracker.

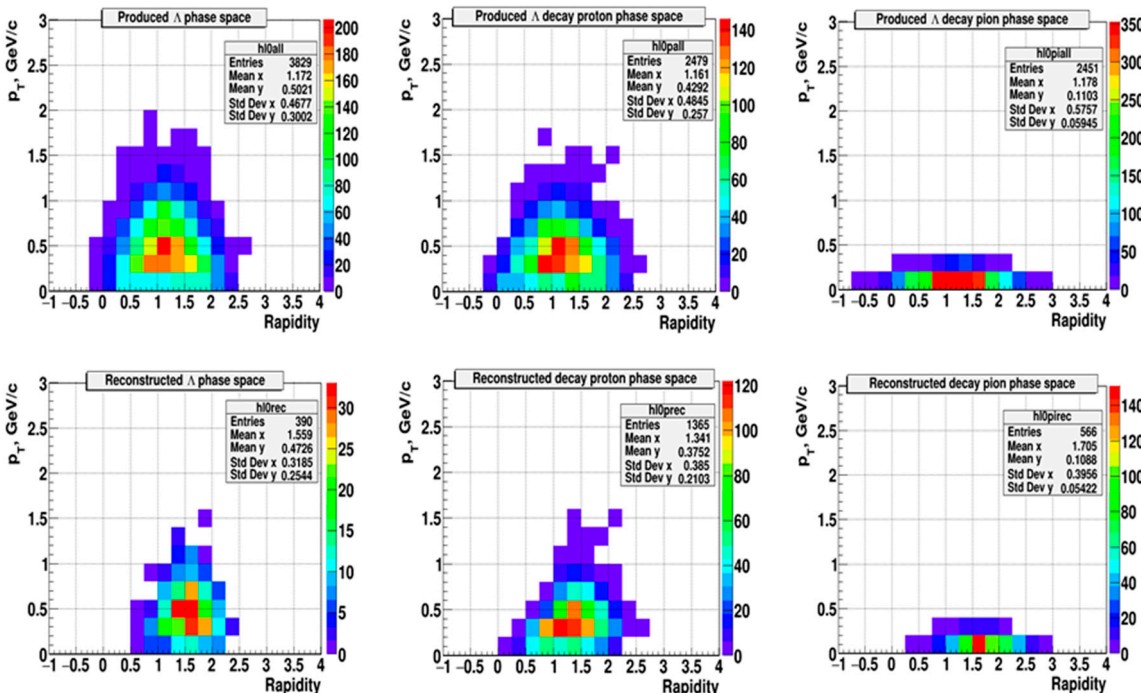

**Figure 7.** Phase space distributions as function of transverse momentum and rapidity for lambdas (**left column**), protons from lambda decays (**center column**), and pions from lambda decays (**right column**), as generated with the DCM-QGSM code for central Au + Au collisions at a beam kinetic energy of 4 A GeV (upper row), and reconstructed based on hits in the hybrid tracking system (lower row).

### 4.3. Expected Particle Yields

The ultimate goal of the BM@N research program devoted to the nuclear EOS would be to measure the excitation functions for the production of $\Lambda$, $\Xi^-$, and $\Omega^-$ hyperons in central Au + Au and central C + C collisions for beam kinetic energies of 2, 3, and 4 A GeV. The multiplicities for $\Lambda$ and $\Xi^-$ hyperons per central Au + Au collision as predicted by a statistical hadronization model [26] are listed in Table 1 for the 3 bombarding energies, together with the expected yields recorded per week. The collision rate is assumed to be 10 kHz. The yield estimate takes into account the reconstruction efficiencies (10% for $\Lambda$, and 1% for $\Xi^-$) and the respective branching ratios. According to these numbers, no $\Xi^-$ hyperon can be reconstructed in 1000 central Au + Au collisions, which have been simulated so far. In order to demonstrate the feasibility of $\Xi^-$ or even $\Omega^-$ hyperon measurements, simulations with much better statistics have to be performed.

**Table 1.** $\Lambda$ and $\Xi^-$ multiplicities per central Au + Au collision according to the statistical hadronization model [26], and the yields measured per week taking into account the reconstruction efficiencies (10% for $\Lambda$, and 1% for $\Xi^-$) and the branching ratios (BR = 0.64 for $\Lambda \to p + \pi^-$ and also for $\Xi^- \to \Lambda + \pi^- \to p + \pi^- + \pi^-$). The kinetic beam energies are indicated, the collision rate is 10 kHz.

| E (A GeV). | M($\Lambda$) | M($\Xi^-$) | $\Lambda$/week | $\Xi^-$/week |
|---|---|---|---|---|
| 2 | 0.15 | $2.6 \times 10^{-4}$ | $5.8 \times 10^7$ | $1.6 \times 10^4$ |
| 3 | 0.75 | $5.7 \times 10^{-3}$ | $2.9 \times 10^8$ | $3.4 \times 10^5$ |
| 4 | 3.8 | 0.11 | $1.5 \times 10^9$ | $6.6 \times 10^6$ |

According to statistical model predictions, also the measurement of hypernuclei should be possible at Nuclotron beam energies [27]. The multiplicity of $^3_\Lambda$H hypernuclei per central Au + Au collisions at 4 A GeV is predicted to be $10^{-2}$, and the multiplicity of both $^4_\Lambda$H and $^4_\Lambda$He hypernuclei is $10^{-3}$. The decay topology of $^3_\Lambda$H $\to$ $^3$He + $\pi^-$ and $^4_\Lambda$H $\to$ $^4$He + $\pi^-$ is similar to the lambda decay. Assuming the hypernuclei reconstruction has the same branching ratio as for the lambdas and a reconstruction efficiency of 1%, a measured yield of about $3 \times 10^4$ $^3_\Lambda$H and $3 \times 10^3$ $^4_\Lambda$H per week of running at a collision rate of 1 kHz should be feasible. The reconstruction of the decay $^4_\Lambda$He $\to$ $^3$He + p + $\pi^-$ should be also possible. These measurements will allow one to determine the lifetime of these hypernuclei with high precision. It should be noted, that the yields mentioned above have been estimated without taking into account a duty factor, neither for the accelerator nor for the experiment

## 5. Conclusions and Outlook

The upgrade of the BM@N experiment with an evacuated beam pipe, a hybrid tracking system comprising of 4 stations of double-sided microstrip silicon sensors and 7 planes of two-coordinate GEM detectors (already partly existing), and a zero degree calorimeter for event characterization, will allow one to investigate Au + Au collisions at Nuclotron energies with interaction rates up to several kHz. These measurements open the possibility, to explore high-density EOS for symmetric matter, and to scout the location of a possible mixed phase of QCD matter. Both the detector developments and the feasibility studies are works in progress. The next simulation steps include particle identification via time-of-flight (TOF), and a substantial increase of the event statistics to be able to demonstrate the feasibility to reconstruct multi-strange hyperons and light hypernuclei. It is worthwhile to note that up to date no multi-strange hyperons have been measured in heavy collision systems in the Nuclotron beam energy range. The goal of the BM@N experiment is to provide the first data on these promising probes of dense QCD matter.

**Author Contributions:** Funding acquisition, H.R.S.; investigation, D.D., J.H., M.K., C.S., and A.S.; methodology, A.M.; resources, Y.M.; software, E.L. and A.Z.; writing—original draft, P.S.; writing—review and editing, P.S.

**Funding:** This research received no external funding.

**Acknowledgments:** The realization and operation of the BM@N experiment is a joint effort of the BM@N Collaboration, which consists of more than 230 persons from 21 institutions and 11 countries. P.S. acknowledges support from the Ministry of Science and Higher Education of the Russian Federation, grant N 3.3380.2017/4.6 and by the National Research Nuclear University MEPhI in the framework of the Russian Academic Excellence Project (contract No. 02.a03.21.0005, 27.08.2013).

**Conflicts of Interest:** The authors declare no conflict of interest.

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
