# Peer review of "Upgrading the Baryonic Matter at the Nuclotron Experiment at NICA for Studies of Dense Nuclear Matter"

_2571-712X, doi:10.3390/particles2040029_

Round 1
Reviewer 1 Report
Referee review of the article:
"Upgrading the BM@N experiment at NICA for studies of dense nuclear matter" by P. Senger et al.
In this article the authors discuss the possibility of studying the phases of dense nuclear matter with the
upgraded Baryon Matter at the Nuclotron (BM@N) experiment. The manuscript presents arguments that support
matter compressions up to four times saturation density, typical of neutron star interiors and temperatures
right above the typical ones of stellar matter in mergers of binaries.
The article properly describes the setup of the BM@N and the energy scan range that is capable of. In addition, the results
of performance simulations of the BM@N hybrid track system are presented. This eventually leads the the expected particle yields
which are discussed in the last section before the conclusions. The manuscript ends with an outlook of the scientific case and
concludes that BM@N will allow for studies of the high density EoS of symmetric matter as well as scouting the location of a mixed
phase of QCD matter.
In my opinion the article is well structured and presents a useful overview of the scientific goals as well
as technical details and scope of the BM@N experiment at NICA. Therefore, I would like to recommend it for publication.
However, I would like just to make a suggestion to be applied in line 50:
"The central densities increase for neutron stars with smaller masses"
to be replaced by
"The central densities increase for neutron star models with smaller maximum mass, i.e., for softer equations of state."
Reviewer 2 Report
The article contains a nice description of the scientifc goals and the set-up of the upgraded BM@N experiment at NICA. It is an important experiment aiming to explore the baryon rich region. One important observable in this region would be the net proton density expected to obtain, based on first principles, a double hump structure in this region (see for example Antoniou et al, Nucl. Phys. A 986, 167 (2019)). This, quite general, behaviour could provide valuable information for the EOS in the baryon rich region as well. The authors could optionally include in their scientific goals such a study. In general, the paper is well written and deserves publication in "Particles".